# Characteristics of Traditional Chinese Medicine Use for Carpal Tunnel Syndrome

**DOI:** 10.3390/ijerph16214086

**Published:** 2019-10-24

**Authors:** Meng-Chuan Tsai, Yu-Hsien Kuo, Chih-Hsin Muo, Li-Wei Chou, Chung-Yen Lu

**Affiliations:** 1Department of Sport and Health Management, Da-Yeh University, Changhua 51591, Taiwan; dykama@mail.dyu.edu.tw; 2Department of Chinese Medicine, China Medical University Hospital, Taichung 40447, Taiwan; u100030033@cmu.edu.tw; 3Management Office for Health Data, China Medical University Hospital, Taichung 40447, Taiwan; b8507006@gmail.com; 4Department of Physical Therapy and Graduate Institute of Rehabilitation Science, China Medical University, Taichung 404, Taiwan; 5Department of Physical Medicine and Rehabilitation, China Medical University Hospital, Taichung 404, Taiwan; 6Department of Physical Medicine and Rehabilitation, Asia University Hospital, Taichung 404, Taiwan; 7Department of Chinese Medicine, China Medical University Hospital, Taipei Branch, Taipei 114, Taiwan

**Keywords:** carpal tunnel syndrome, traditional Chinese medicine, occupational disease

## Abstract

Carpal tunnel syndrome (CTS) is a common musculoskeletal disorder and an occupational disease caused by repeated exercise or overuse of the hand. We investigated the characteristics of traditional Chinese medicine (TCM) use by practitioners in CTS patients, including demographic variables, socioeconomic status, previous medical conditions, health care use, and hospital characteristics for TCM health care. This cross-sectional study identified 25,965 patients newly diagnosed with CTS based on the first medical diagnosis recorded between 1999 and 2013 in the nationwide representative insurance database of Taiwan. The date of initial CTS diagnosis in outpatient data was defined as the index date, and four patients were excluded because of missing gender-related information. Patients who used TCM care as the first option at their diagnosis were classified as TCM users (*n* = 677; 2.61%), and all others were TCM non-users (*n* = 25,288; 97.4%). In the all variables-adjusted model, female patients had an adjusted odds ratio (OR; 95% CI) of TCM use of 1.35 (1.11–1.66). National Health Insurance (NHI) registration was associated with higher odds ratios of TCM use in central, southern, and eastern Taiwan than in northern Taiwan (ORs = 1.43, 1.86, and 1.82, respectively). NHI registration was associated with higher odds ratios of TCM use in rural cities than in urban cities (OR (95% CI) = 1.33 (1.02–1.72)). The TCM group had a 20% less likelihood of exhibiting symptoms, signs, and ill-defined conditions and injury and poisoning. The TCM group had a 56% lower likelihood of having diseases of the musculoskeletal system and connective tissue. Multi-level model outcomes were similar to the results of the all variables-adjusted model, except for the NHI registration outcome in rural and urban cities (OR [95% CI] = 1.33 [0.98–1.81]). Significant associations between the number of TCM visits and TCM use were observed in all logistic regression models. The study presented key demographic characteristics, health care use, and medical conditions associated with TCM use for CTS. Previous experience of TCM use may affect the use of TCM for CTS treatment. This information provides a reference for the allocations of relevant medical resources and health care providers.

## 1. Introduction

Carpal tunnel syndrome (CTS) is an occupational disease of the musculoskeletal system caused by repetitive movements or overuse of the hand [1]. CTS cases are reported to be associated with the following occupational factors: forceful hand exertion, repeated trauma of the hand, repeated movements of the wrist, wrists in awkward postures, or vibration [1,2]. This condition is often prevalent in industries involving production workers, material-moving workers, and office and administrative support workers [2].

Several longitudinal and cross-sectional studies have investigated the epidemiology of CTS. A longitudinal study reported that the incidence proportion of CTS was 2.0% in women cashiers [3]. During 2007–2014 in California, the overall incidence rate of CTS among full-time equivalent workers was 6.3 cases per 10,000 workers [2]. A cross-sectional study of the meat-processing industry in Northern Italy reported the diagnosis of CTS with an incidence of 11.3/1000 person-years [1]. Although the incidence of CTS differs considerably among industries, comorbidities caused by CTS cannot be ignored, as 40% of new CTS cases have comorbidities. For example, overweight/obesity and hypothyroidism are the most frequent comorbidities [3].

The pathology of CTS involves nerve compression, and expected symptoms include pain, sensorimotor abnormalities, and loss of hand function [4]. Most patients with mild and moderate forms of CTS experience pain, have to undergo prolonged treatment, and are distressed about the possibility of recurrence. The performance in the two-point discrimination sense, kinesthetic differentiation of strength, and kinesthetic differentiation of movement are poor [5]. Patients with severe CTS may experience severe pain or numbness, impaired hand function, muscular atrophy, and electrical diagnosis confirming nerve damage. Patient-reported symptom severity in CTS makes work more difficult and affects quality of life [6,7].

Pain from CTS can be relieved through nonsteroidal anti-inflammatory drugs (NSAIDs), physical therapy, and surgery. However, NSAIDs may cause kidney damage, ultrasounds may result in nerve damage, and surgery may injure normal tissues. Therefore, seeking alternative therapy for CTS is necessary. Seradge et al. published in 2002 an investigation of the relationship between CTS and alternative medicine treatment and presented alternative medicine treatments that can significantly reduce human suffering, medical costs, lost work time, and socioeconomic problems [8].

Traditional Chinese medicine (TCM) is thousands of years old and is a crucial category of complementary and alternative medical treatments. TCM treatment has been frequently practiced in many countries. Multi-racial Singapore has a high prevalence rate (76%) of complementary and alternative medicine [9]. Over 66.67% of older Chinese immigrants in Canada use TCM in combination with Western health services [10]. In Hong Kong, among patients who claim to have medical benefits or insurance policies, 14.5% are covered for TCM. Those belonging to higher socioeconomic classes have emerged as a new class of TCM users, rates of TCM use are low among institutionalized elderly persons, and elderly persons are less likely to be covered by TCM insurance [11]. In Taiwan, National Health Insurance (NHI) payment program benefits include TCM. By the end of 2001, 28.4% among valid beneficiaries of the NHI used TCM [12]. However, the main health care benefit for TCM outpatient services comes from the universal health insurance program, in which the total amount of medical expenses assigned for TCM accounted for approximately 3%. We wondered whether such distribution of payments was sufficient to provide safe and effective medical services. We investigated factors associated with TCM use, including demographics, socioeconomic status, medical conditions, and health care use, among populations with CTS by using nationwide representative claims data. This information contributed to medical resource allocation and policy development for integrative health services concerning CTS populations.

## 2. Materials and Methods

### 2.1. Data Source

The National Health Insurance Research Database (NHIRD) is a computer database containing encrypted claims records of people insured in Taiwan’s NHI program. The NHI has been operational since March 1995, providing medical insurance for more than 99% of Taiwan’s total population and contracting more than 90% of hospitals and clinics. In this analysis, we used NHIRD’s sub-dataset, the Longitudinal Health Insurance Database that contains individual claims data of a population of 1 million people randomly selected from the 23.75 million insured people between 1996 and 2000. The random number function, Sun WorkShop C 5.0, was used to generate random numbers, and linear congruential random number generation was used to select 1.1 million random numbers, from which repetitive random numbers (approximately 20,000–30,000) were removed. Finally, the remaining numbers were divided 20 times. Each time, 50,000 pieces of data were taken out in sequence, and a total of 1 million random numbers were obtained. Patients in this group were similar to NHI registrants in terms of gender and age distributions. The database consists of inpatient and outpatient expenditures and orders and a beneficiary registry that together provide health care data, outpatient visits and hospitalization dates, medical diagnostic information, medical services received, and limited demographic characteristics [13,14]. The Institutional Review Board of China Medical University Hospital (CMUH104-REC2-115(CR-1)) approved this study.

### 2.2. Study Sample

We conducted a cross-sectional study using data from the Longitudinal Health Insurance Database. The flowchart of patient selection is displayed in Figure 1. We identified 25,965 patients newly diagnosed with CTS (International Classification of Diseases, Ninth Edition, Clinical Revision [ICD-9-CM] code 354.0) based on the diagnosis at the first medical visit recorded between 1999 and 2013. We chose this time frame to allow for the exclusion of medical history assessments for at least three years, as the computerized claims data have been available since 1996. The date of the first outpatient visit was defined as the index date. Patients were excluded if they lacked gender-related information. Patients who used TCM services at the time of initial diagnosis were classified as TCM users and the others as TCM non-users.

### 2.3. Patient Characteristics

Given that influencing factors and confounders could be related to TCM use in CTS patients, the characteristics discussed in this study included demographic variables, socioeconomic status, previous medical conditions, health care utilization, and hospital characteristics for TCM health care. Demographic variables included age, gender, and geographic area of patients’ NHI registry, which was typically the location of beneficiaries’ employment or place of residence of parents on the index date. Socioeconomic status included monthly income, occupation, and level of urbanization. According to Taiwan’s total judicial budgetary, accounting, and statistical classification scheme, the urbanization level was divided into three levels: urban areas, satellite cities or towns, and rural areas [15]. Monthly income was divided into three levels: <15,840, 15,840–25,000, and >25,000 (NTD). Occupation was also divided into three levels: white collar, blue collar, and other jobs. We assessed the use of inpatient and outpatient care and medical conditions by searching for inpatient and outpatient claims records within one year before the index date. The comorbidities considered in our analysis included infectious and parasitic diseases (ICD-9-CM 001–139); neoplasms (ICD-9-CM 140–239); endocrine, nutritional, blood, metabolic diseases, and immunity disorders (ICD-9-CM 240–289); mental disorders; diseases of the nervous system and sensory organs (ICD-9-CM 290–389); diseases of the circulatory system (ICD-9-CM 390–459); diseases of the respiratory system (ICD-9-CM 560–519); diseases of the digestive system (ICD-9-CM 520–579); diseases of the genitourinary system (ICD-9-CM 580–677); diseases of the skin and subcutaneous tissue (ICD-9-CM 680–709); diseases of the musculoskeletal system and connective tissue (ICD-9-CM 710–739); symptoms, signs, and ill-defined conditions (ICD-9-CM 780–799); injury and poisoning (ICD-9-CM 800–999); supplementary classification (ICD-9-CM V01–V82 and E800–E999); and other conditions (ICD-9-CM 740–759 and 760–779). We collected information on the location and certification level of the hospitals where the patients received TCM health care for CTS [16].

### 2.4. Statistical Analysis

Data analysis initially used descriptive measures to account for patient characteristics, including previous medical conditions, demographic variables, socioeconomic status, health care use, and hospital characteristics for TCM health care. These variables were compared between patients with CTS who used and did not use TCM as their first option by using chi-square tests, and mean age was determined using the independent sample *t* test. We divided the area of each patient’s NHI registration unit into four geographic regions (northern, central, southern, and eastern), and the urbanization status of each region was divided into three levels (urban, satellite, and rural). The levels of certification for hospitals were divided between medical centers, district hospitals, local hospitals, and clinics.

We performed a series of logistic regression models that included variables either individually or simultaneously to calculate odds ratios (ORs) and 95% CIs and to assess whether each of these variables was related to TCM use. A crude model estimated the unadjusted association between each variable. An age- and sex-adjusted model was adjusted for age and sex. A selected variables-adjusted model was adjusted for variables that were statistically significantly associated with TCM use in the crude model. A model adjusted for all variables and a multilevel model were adjusted for all variables. All statistical analyses were performed using SAS v.9.3 (SAS Institute Inc., Carey, NC, USA). Bilateral probability values <0.05 were considered statistically significant.

## 3. Results

During 1999–2013, 25,965 patients had their first outpatient clinic visit with diagnoses of CTS, which was included in the data analysis (Figure 1). Table 1 lists the characteristic data of patients who used TCM and those who did not use TCM as their preferred medical care for CTS. Of all patients, only 677 (2.61%) received TCM care at their first clinic visit. The majority of patients were aged between 46 and 65 years (51.5%). Women accounted for 71.3% of all patients. The majority of patients were had NHI registration in northern Taiwan (45.4%) and in urban cities (61.1%). In the previous year, most patients had diseases of the musculoskeletal system and connective tissue (66.1%).

Patients using TCM were younger than those who did not use TCM (aged 46.3 vs. 50.5 years, *p* < 0.0001). The proportion of female users of Chinese medicine was higher than that of female non-users (78.7% vs. 71.1%, *p* < 0.0001). People with NHI registration in southern Taiwan and in rural cities were more likely to use TCM for CTS treatment (39.4% vs. 29.4%, *p* < 0.001; 12.4% vs. 9.02%, *p* < 0.001, respectively). TCM users were less likely to have neoplasm diseases (*p* = 0.02); endocrine, nutritional, blood, and metabolic diseases; immunity disorders (*p* < 0.0001); mental disorders; diseases of the nervous system and sensory organs (*p* = 0.0005); diseases of the circulatory system (*p* < 0.0001); and diseases of the musculoskeletal system and connective tissue (*p* < 0.0001) within one year before being diagnosed with CTS than TCM non-users. The TCM group had a smaller number of outpatient clinic visits (median 20 vs. 23, *p* < 0.0001) and a larger number of TCM outpatient clinic visits (median four vs. zero, *p* < 0.0001).

Table 2 presents characteristics of the hospitals in which patients received medical care for CTS after their diagnosis. In hospitals located in northern Taiwan, 44.4% of patients received medical care. In hospitals in eastern Taiwan, only 4.86% of patients received medical care. In district hospitals, 32.2% of patients were likely to receive medical care. TCM users were less willing to receive treatment in hospitals in northern Taiwan than TCM non-users (29.3% vs. 44.8%, *p* < 0.0001). Most TCM users received medical care in clinics (83.9%), which was over three times that of TCM non-users (24.6%, *p* < 0.0001).

Table 3 presents the relationship between the ORs of patients using TCM for CTS and their demographic characteristics, socioeconomic status, health care use, and medical conditions within a year before their diagnosis. In the crude logistic regression model, among CTS patients, TCM use was associated with aged 46–65, aged >65 years, female, having NHI registrations in central, southern, and eastern Taiwan, living in rural cities, having month incomes (NTD) of <15,840, having neoplasm diseases, having endocrine, nutritional, blood, and metabolic diseases, having immunity and mental disorders, having diseases of nervous system and sensory organs, circulatory system, having musculoskeletal system and connective tissue, having various outpatient visits with the numbers between Q2 and Q3 and ≥Q3, and having inpatient visits with the numbers ≥Q3.

TCM use for CTS treatment was associated with having NHI registration in central, southern, and eastern Taiwan; living in rural cities; having neoplasms diseases; having endocrine, nutritional, blood, and metabolic diseases; having immunity and mental disorders; having diseases of the nervous system and sensory organs, circulatory system, skin and subcutaneous tissue, and musculoskeletal system and connective tissue; and having various outpatient visits in ≥Q3 in the age- and sex-adjusted model.

However, no significant association was observed between TCM use, demographics, socioeconomic status, and medical conditions in CTS patients, except for being female; having NHI registration in central, southern, and eastern Taiwan; living in rural cities; having diseases of the musculoskeletal system and connective tissue; and having various outpatient visits during Q1–Q2, Q2–Q3, and ≥Q3 in the selected variables-adjusted model.

In the all variables-adjusted model, the adjusted OR (95% CI) of TCM use was 1.35 (1.11–1.66) in female patients as compared with that in male patients. NHI registrations were associated with higher odds of TCM use in central, southern, and eastern Taiwan than in northern Taiwan (ORs = 1.43, 1.86, and 1.82, respectively). NHI registration was associated with higher odds of TCM use in rural cities than in urban cities (OR 95% CI) = 1.33(1.02–1.72)). The TCM group had an approximately 20% lesser likelihood of having symptoms, signs, and ill-defined conditions and injury and poisoning, with a 56% reduced likelihood of having diseases of the musculoskeletal system and connective tissue. Various outpatient visits were associated with less likelihood of TCM use during Q2–Q3 and ≥Q3 than in <Q1 (ORs = 0.68 and 0.62, respectively).

The results of the multilevel model were similar to those of the all variables-adjusted model except for the urbanization comparison. In the multilevel model, NHI registration in rural cities was not associated with TCM use compared with that in urban cities (OR (95% CI) = 1.33 (0.98–1.81)). The associations were observed between TCM use and number of TCM visits in all logistic regression models.

## 4. Discussion

Unlike other diseases, CTS is closely related to chronic occupational injuries [1,2]. In this population-based study in Taiwan, half (51.5%) of the CTS population was aged between 46 and 65 years. A study conducted in Canada over 15 years by Jackson et al. reported a similar result, in which 47% of the CTS population was aged 45–64 years among the working population [2].

Geographic region and urbanization level of NHI registration location, particularly in southern and urban Taiwan, were associated with the initiation of TCM use, which differed from other studies investigating other diseases. For example, pediatric TCM users with dislocations and sprains and strains were mainly living in northern Taiwan and satellite cities, respectively, [17] and most other diseases were associated with TCM use in central Taiwan [18]. That most patients using TCM for CTS treatment lived in rural areas was a different outcome from other studies investigating other diseases [19]. However, 71.3% of the CTS population in Taiwan being female was similar to populations in other studies [18,19,20]. In total, 83.9% of TCM users received care at local clinics, which accounted for approximately 80% of medical care services in Taiwan.

Unlike other studies, this study revealed that three systemic diseases had negative correlations with TCM use for CTS. Systemic diseases of the musculoskeletal system and connective tissue were the most encountered diseases in the TCM population. However, in this study, diseases of the musculoskeletal system and connective tissue showed a negative correlation with TCM use for CTS, and this merits further discussion.

Our observations revealed no significant association between the number of inpatient visits and TCM users with initial CTS. A negative correlation was observed between the number of outpatient visits and TCM users with initial CTS. The number of previous TCM visits had a strong association with TCM users with initial CTS that reflected how past experience with TCM could affect the initial treatment chosen for CTS.

We used the claims database in this study. Thus, our findings must be interpreted with their limitations. First, the NHI reimbursed only ambulatory care services of TCM, so this study did not include patients who received therapy solely with inpatient services. Although CTS is unlikely to become serious enough to require hospitalization, the observations in this study are not generalizable to CTS patients who receive treatment during hospital admissions. Second, TCM services provided by health care institutions not contracted by the NHI were not available in the claims database. TCM use may be underestimated for the TCM user group and misclassified into the nonuse group for patients seeking care outside the NHI-contracted network. However, most medical care institutions are NHI-contracted. As of 2013, the contract rate was 92.9% of TCM hospitals and 92.4% of TCM clinics [21]. Finally, as the claims database does not contain information including smoking, alcohol use, betel nut use, exercise, and dietary patterns, we were unable to assess detailed lifestyle and behavioral factors.

## 5. Conclusions

In this population-based study, 2.61% of patients newly diagnosed with CTS initiated TCM treatment after their diagnosis. Among CTS patients, being female; having NHI registration in central, southern, and eastern Taiwan; and living in areas with a rural urbanization level were associated with the initiation of TCM use. The use of TCM care service for CTS was associated with previous experience of TCM usage. TCM outpatient services were mainly provided by local clinics. These observations can aid resource allocation for health care system policy regarding CTS populations. Further studies are required to evaluate the outcomes of TCM health care for CTS.

## Figures and Tables

**Figure 1 ijerph-16-04086-f001:**
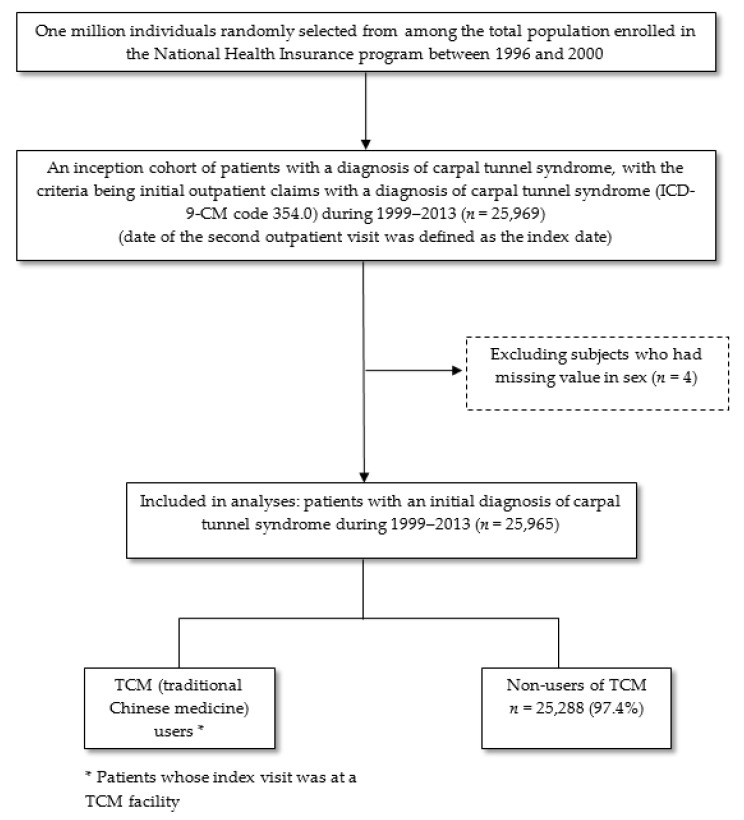
Flowchart of study sample selection.

**Table 1 ijerph-16-04086-t001:** Baseline characteristics of patients with and without TCM use as the preferred medical care for carpal tunnel syndrome at diagnosis.

Characteristics	Total	Non-Users of TCM	TCM Users	*p*
	*N* = 25,965	*N* = 25,288 (97.4%)	*N* = 677 (2.61%)	
Age at diagnosis, yr, *n* (%)				<0.0001
<18	54 (0.21)	52 (0.21)	2 (0.30)	
18–30	1644 (6.33)	1580 (6.25)	64 (9.45)	
31–45	7689 (29.6)	7424 (29.4)	265 (39.1)	
46–65	13,360 (51.5)	13,058 (51.6)	302 (44.6)	
>65	3218 (12.4)	3174 (12.6)	44 (6.50)	
Mean(SD)	50.4 (12.9)	50.5 (1.29)	46.3 (12.2)	<0.0001
Female, *n* (%)	18,505 (71.3)	17,972 (71.1)	533 (78.7)	<0.0001
Geographic region of registration units for NHI, *n* (%)				<0.0001
Northern	11,793 (45.4)	11,573 (45.8)	220 (32.5)	
Central	5148 (19.8)	4998 (19.8)	150 (22.2)	
Southern	7692 (29.6)	7425 (29.4)	267 (39.4)	
Eastern	1332 (5.12)	1292 (5.11)	40 (5.91)	
Urbanization level of registration units for NHI *, *n* (%)				0.01
Urban	15,856 (61.1)	15,459 (61.1)	397 (58.6)	
Satellite	7743 (29.8)	7547 (29.8)	196 (29.0)	
Rural	2366 (9.11)	2282 (9.02)	84 (12.4)	
Previous medical conditions ^†^, *n* (%)				
Month income (NTD), *n* (%)				0.02
<15,840	8429 (32.5)	8176 (32.3)	253 (37.4)	
15,840–25,000	11,656 (44.9)	11,369 (45.0)	287 (42.4)	
>25,000	5880 (22.7)	5743 (22.7)	137 (20.2)	
Occupation, *n* (%)				0.99
White collar	13,099 (50.5)	12,758 (50.5)	341 (50.4)	
Blue collar	11,022 (42.5)	10,734 (42.5)	288 (42.5)	
Others	1844 (7.10)	1796 (7.10)	48 (7.09)	
Infectious and parasitic diseases	3449 (13.3)	3366 (13.3)	83 (12.3)	0.43
Neoplasms	2067 (7.96)	2030 (8.03)	37 (5.47)	0.02
Endocrine, nutritional, blood and metabolic diseases, and immunity disorders	6504 (25.1)	6386 (25.3)	118 (17.4)	<0.0001
Mental disorders, diseases of the nervous system and sense organs	16,042 (61.8)	15,667 (62.0)	375 (55.4)	0.0005
Diseases of the circulatory system	7852 (30.2)	7711 (30.5)	141 (20.8)	<0.0001
Diseases of the respiratory system	17,108 (65.9)	17,108 (67.7)	457 (67.5)	0.93
Diseases of the digestive system	15,854 (61.1)	15,444 (61.1)	410 (60.6)	0.79
Diseases of the genitourinary system	8745 (33.7)	8520 (33.7)	225 (33.2)	0.80
Diseases of the skin and subcutaneous tissue	6780 (26.1)	6625 (26.2)	155 (22.9)	0.054
Diseases of the musculoskeletal system and connective tissue	17,174 (66.1)	16,838 (66.6)	336 (49.6)	<0.0001
Symptoms, signs, and ill-defined conditions	11,786 (45.4)	11,493 (45.5)	293 (43.3)	0.26
Injury and poisoning	8435 (32.5)	8230 (32.6)	205 (30.3)	0.21
Supplementary classification	4456 (17.2)	4348 (17.2)	108 (16.0)	0.40
Others	215 (0.83)	210 (0.83)	5 (0.74)	0.79
Health care utilization ^†^, *n* (%)				
Number of outpatient visit				<0.0001
Q1 (25%)	13	13	12	
Q2 (50%)	23	23	20	
Q3 (75%)	37	37	33	
Median (interquartile range)	23 (24)	23 (24)	20 (21)	
Number of inpatient visit				0.009
Q1 (25%)	0	0	0	
Q2 (50%)	0	0	0	
Q3 (75%)	0	0	0	
Median (interquartile range)	0 (0)	0 (0)	0 (0)	
Number of TCM visit				<0.0001
Q1 (25%)	0	0	2	
Q2 (50%)	0	0	4	
Q3 (75%)	3	3	8	
Median (interquartile range)	0 (3)	0 (3)	4 (6)	

Abbreviations: TCM, traditional Chinese medicine. Values are number of patients and percentages unless otherwise indicated. * Directorate-General Budget, Accounting and Statistics; National statistics of regional standard classification data; Taipei: Accounting and Statistics; 1993. ^†^ Defined by searching claims data within one year before index date. Comorbidities were considered to be present if the diagnosis codes were recorded on at least one inpatient claim or two outpatient claims. *n*, number of patients with carpal tunnel syndrome.

**Table 2 ijerph-16-04086-t002:** Characteristics of hospitals where patients received medical care after their diagnosis of carpal tunnel syndrome.

Characteristics	Total	Non-Users of TCM	TCM Users	*p*
	*N* = 25,965	*N* = 25,288	*N* = 677	
Locations of hospital where subjects received the diagnosis, *n* (%)				<0.0001
Northern	11,517 (44.4)	11,319 (44.8)	198 (29.3)	
Central	5337 (20.6)	5184 (20.5)	153 (22.6)	
Southern	7849 (30.2)	7568 (29.9)	281 (41.5)	
Eastern	1262 (4.9)	1217 (4.8)	45 (6.7)	
Accreditation level of hospital where subjects received the diagnosis, *n* (%)				<0.0001
Medical center	5097 (19.6)	5067 (20.0)	30 (4.4)	
District hospital	8367 (32.2)	8314 (32.9)	53 (7.8)	
Local hospital	5707 (22.0)	5681 (22.5)	26 (3.8)	
Clinics and others	6794 (26.2)	6226 (24.6)	568 (83.9)	

**Table 3 ijerph-16-04086-t003:** Demographic characteristics, health care use, and medical conditions within a year before diagnosis were associated with using TCM as the preferred medical care in patients with carpal tunnel syndrome.

Characteristics	Odds Ratio (95% Confidence Interval) ^†^				
	Crude Model	Age and Sex Adjusted Model	Selected Variables Adjusted Model *	All Variables Adjusted Model	Multilevel Model
Age at diagnosis, year (vs. ≤30)					
31–45	0.90 (0.68–1.21)	-	1.08 (0.79–1.47)	1.07 (0.78–1.47)	1.05 (0.76–1.44)
46–65	0.58 (0.44–0.78)	-	0.90 (0.65–1.24)	0.87 (0.63–1.22)	0.86 (0.62–1.20)
>65	0.35 (0.23–0.52)	-	0.78 (0.51–1.22)	0.73 (0.47–1.15)	0.73 (0.46–1.15)
Female (vs. Male)	1.51 (1.25–1.82)	-	1.33 (1.10–1.62)	1.35 (1.11–1.66)	1.36 (1.11–1.66)
Geographic region of registration units for NHI (vs. northern)					
Central	1.58 (1.28–1.95)	1.56 (1.26–1.92)	1.41 (1.12–1.77)	1.43 (1.13–1.79)	1.36 (1.01–1.82)
Southern	1.89 (1.58–2.27)	1.90 (1.58–2.27)	1.86 (1.54–2.26)	1.86 (1.53–2.26)	1.91 (1.47–2.48)
Eastern	1.63 (1.16–2.29)	1.64 (1.16–2.30)	1.85 (1.29–2.65)	1.82 (1.26–2.61)	1.84 (1.19–2.84)
Urbanization level of registration units for NHI * (vs. urban)					
Satellite	1.01 (0.85–1.20)	1.01 (0.85–1.20)	0.89 (0.74–1.07)	0.90 (0.75–1.08)	0.93 (0.74–1.17)
Rural	1.43 (1.13–1.82)	1.47 (1.15–1.86)	1.32 (1.02–1.71)	1.33 (1.02–1.72)	1.33 (0.98–1.81)
Month income (vs. >25,000)					
<15,840	1.30 (1.05–1.60)	1.09 (0.88–1.36)	1.09 (0.87–1.37)	1.09 (0.86–1.39)	1.12 (0.88–1.43)
15,840–25,000	1.06 (0.86–1.30)	1.03 (0.84–1.27)	0.98 (0.79–1.21)	0.98 (0.77–1.23)	1.00 (0.79–1.27)
Occupation (vs. white collar)					
Blue collar	1.00 (0.86–1.18)	1.05 (0.89–1.23)	-	1.02 (0.84–1.24)	0.99 (0.81–1.22)
Others	1.00 (0.74–1.36)	1.11 (0.81–1.50)	-	1.06 (0.76–1.47)	1.09 (0.85–1.39)
Common conditions of pediatric patients in outpatient settings ^†^, (yes vs. no)					
Infectious and parasitic diseases	0.91 (0.72–1.15)	0.95 (0.75–1.20)	-	1.08 (0.84–1.38)	1.09 (0.85–1.39)
Neoplasms	0.66 (0.47–0.93)	0.68 (0.48–0.95)	0.76 (0.54–1.07)	0.76 (0.54–1.08)	0.76 (0.54–1.08)
Endocrine, nutritional, blood and metabolic diseases, and immunity disorders	0.63 (0.51–0.76)	0.77 (0.62–0.94)	0.91 (0.73–1.13)	0.90 (0.72–1.12)	0.91 (0.73–1.13)
Mental disorders, diseases of the nervous system and sense organs	0.76 (0.65–0.89)	0.83 (0.71–0.97)	0.86 (0.72–1.01)	0.86 (0.72–1.01)	0.85 (0.72–1.01)
Diseases of the circulatory system	0.60 (0.50–0.72)	0.80 (0.65–0.97)	0.95 (0.76–1.18)	0.95 (0.76–1.18)	0.93 (0.75–1.16)
Diseases of the respiratory system	0.99 (0.84–1.17)	0.95 (0.81–1.12)	-	0.99 (0.83–1.19)	1.00 (0.83–1.19)
Diseases of the digestive system	0.98 (0.84–1.14)	1.03 (0.88–1.21)	-	1.11 (0.93–1.31)	1.09 (0.92–1.30)
Diseases of the genitourinary system	0.98 (0.83–1.15)	0.88 (0.74–1.04)	-	0.91 (0.76–1.10)	0.91 (0.76–1.10)
Diseases of the skin and subcutaneous tissue	0.84 (0.70–1.00)	0.83 (0.69–0.99)	-	0.87 (0.72–1.06)	0.87 (0.72–1.06)
Diseases of the musculoskeletal system and connective tissue	0.49 (0.42–0.58)	0.55 (0.47–0.64)	0.44 (0.37–0.52)	0.44 (0.37–0.52)	0.44 (0.37–0.53)
Symptoms, signs, and ill-defined conditions	0.92 (0.79–1.07)	0.96 (0.82–1.12)	-	0.81 (0.68–0.96)	0.81 (0.68–0.97)
Injury and poisoning	0.90 (0.76–1.06)	0.92 (0.78–1.09)	-	0.79 (0.66–0.95)	0.80 (0.64–0.95)
Supplementary classification	0.91 (0.74–1.13)	1.01 (0.82–1.25)	-	1.13 (0.90–1.42)	1.12 (0.89–1.41)
Others	0.89 (0.37–2.16)	0.95 (0.39–2.31)	-	0.93 (0.37–2.30)	0.93 (0.37–2.32)
Health care utilization ^†^					
Number of outpatient visit (vs. <Q1)					
Q1–Q2	0.96 (0.79–1.18)	0.98 (0.80–1.21)	0.80 (0.64–1.00)	0.84 (0.66–1.07)	0.85 (0.67–1.08)
Q2–Q3	0.79 (0.64–0.97)	0.84 (0.68–1.05)	0.62 (0.48–0.79)	0.68 (0.51–0.91)	0.69 (0.52–0.93)
≥Q3	0.63 (0.50–0.79)	0.78 (0.62–0.99)	0.53 (0.39–0.71)	0.62 (0.43–0.89)	0.62 (0.43–0.89)
Number of inpatient visit (vs. no)	0.72 (0.55–0.93)	0.79 (0.60–1.02)	0.95 (0.72–1.25)	0.97 (0.73–1.28)	0.97 (0.73–1.28)
Number of TCM visit (vs. <Q3)	3.82 (3.27–4.47)	3.67 (3.14–4.29)	5.41 (4.54–6.46)	5.80 (4.84–6.96)	5.79 (4.83–6.95)

Abbreviations: TCM, traditional Chinese medicine. * Model was adjusted for variables statistically significantly associated with TCM use in the univariate analysis. ^†^ Logistic regression model.

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
