# Peer review of "Characteristics of Traditional Chinese Medicine Use for Carpal Tunnel Syndrome"

_ijerph, 2019, doi:10.3390/ijerph16214086_

Round 1

Reviewer 1 Report

Sir,

thank you for the opportunity to review this very interesting paper about the demographics of patients referring to traditional chinese medicine for the treatment of CTS. Because of its ancient history, TCM has deep roots in the Eastern World, including both Taiwan and PRC, deserving appropriate assessment by Western Medicine at least in terms of characteristics of people preferring traditional approach over an EBM western-like approach. 

In facts, Authors have specifically focused on the demographics of TCM-referring patients, avoiding any further analysis on the causes of such preference.

I've no concerns about methodological issues, and the soundness of data - mainly descriptive ones, seems appropriate.

However, I still have some remarks and requests for the Authors.

First at all, I think that your data presentation lack of detail. It would be of significant interest provide some data about socioeconomic status of patients, at least about TCM patients.

Second, the description of the modelling for the regression analysis should be improved: in table 3 you report three OR values (Adjusted for age
and sex, Adjusted for selected variables, Adjusted for all variables), but both rationale and methodology applied for identify "selected variables" are not clarified across the text.

Third (and in fact my main concern): the text absolutely requires an extensive English editing. Not only several typos are extensively scattered across the text (e.g. in very first sentence "Carpal Tunnel Syndrome (CTS) is an occupational disease of musculoskeletal condition that
caused by repetitive movement or overuse of the hand..." lacks the verb "is" after "that", and so on), but many sentences appear directly translated Chinese --> English, without the necessary refinement and adaptations.

However, I think that after appropriate amendment your paper may deserve a full publication on IJERPH.

Author Response

Responses to reviewer’s comments

Ref.: MS ID#: ijerph-591433

MS TITLE:

The authors thank the reviewers for their helpful and thoughtful comments and suggestions. In the document below we have listed our responses point by point. Reviewer’s comments are numbered and in Cambria font. Responses follow in Calibri font.

Response to Reviewer

First at all, I think that your data presentation lack of detail. It would be of significant interest provide some data about socioeconomic status of patients, at least about TCM patients.

Reply: Thank you for your help. Your suggestions really help us improve the manuscript. The data about socioeconomic variables included month income and occupation on the index date of each study subject were analyzed and presented in the manuscript and Table1 as well as 3.

(Please see page 4, lines 133-138 in the revised manuscript).

Second, the description of the modelling for the regression analysis should be improved: in table 3 you report three OR values (Adjusted for age and sex, Adjusted for selected variables, Adjusted for all variables), but both rationale and methodology applied for identify "selected variables" are not clarified across the text.

Reply: Thank you for the comment and suggestion. The methodology applied for identify "selected variables" are clarified in the manuscript (Please see page 5, lines 164-168 in the revised manuscript).

Third (and in fact my main concern): the text absolutely requires an extensive English editing. Not only several typos are extensively scattered across the text (e.g. in very first sentence "Carpal Tunnel Syndrome (CTS) is an occupational disease of musculoskeletal condition that caused by repetitive movement or overuse of the hand..." lacks the verb "is" after "that", and so on), but many sentences appear directly translated Chinese --> English, without the necessary refinement and adaptations.

Reply: Thank you for the comment and suggestion. The revised manuscript is re-written by someone with a better command of the English language.

Reviewer 2 Report

This paper describes a large cross-sectional database study on factors associated with TCM use for Carpal Tunnel Syndrome (CTS).

Below are my comments regarding the conduct and reporting of this study.

Abstract.

Please state a clear study objective. Which characteristics did you study?

Readers should be able to read the abstract on his own. So you shouldn’t refer to model 4.

In general, the manuscript needs to be checked for English language usage by a native speaker.

Introduction.

The study authors fail to make clear what the rationale is of this study. Why is it important to investigate factors that are associated with TCM use in CTS? Is there any evidence in favor of TCM as a CTS treatment? Is it desirable that many people use TCM for the treatment of CTS or should they ideally prefer other more evidence-based treatments?

The statement ‘The purpose of this study is to clarify the demographics and patterns of TCM used for CTS through the National Health Insurance (NHI) Database in Taiwan.’ is too vague. What do you mean by clarifying demographics and patterns, which factors would you like to study and why?

Methods.

You are recommended to use the STROBE guidelines for reporting cross-sectional studies.

What is meant by epidemiological history assessment?

A large number of demographic variables and comorbidity variables have been listed. Are these variables all variables of interest or are some of them considered confounders? Please clarify?

I can imagine that there might be some clustering of data. For example, in urban areas and geographic regions. Did you consider multi-level logistic regression analyses or did you have reasons to stick to normal logistic regression analyses?

Discussion and conclusions

The female lead to a higher probability of CTS is due to the long-term fixed posture of the work patterns.’ This is speculative. This wasn’t investigated in this study. At least some some solid references are need to substantiate this statement. Otherwise, it should be removed.

Although the number of people living in urban areas use more proportion of TCM, however, people living in rural areas have a relatively high probability of using TCM to treat CTS.’ I don’t understand this. This is a contradiction.

Author Response

Responses to reviewer’s comments

Ref.: MS ID#: ijerph-591433

MS TITLE:

The authors thank the reviewers for their helpful and thoughtful comments and suggestions. In the document below we have listed our responses point by point. Reviewer’s comments are numbered and in Cambria font. Responses follow in Calibri font.

Response to Reviewer

Please state a clear study objective. Which characteristics did you study? Readers should be able to read the abstract on his own. So you shouldn’t refer to model 4. In general, the manuscript needs to be checked for English language usage by a native speaker..

Reply: Thank you for your help. Your suggestions really help us improve the manuscript. We state the study objective that “In order to understand the characteristics of the use of traditional Chinese medicine (TCM) practitioners in CTS patients, we investigated the characteristics related to TCM use, including demographic variables, socioeconomic status, previous medical conditions, health care utilization, and hospital characteristics for TCM health care.” in the revised manuscript (Please see page 1, lines 22-25 in the revised manuscript). We also delete the words “(model 4)” from the abstract in the revised manuscript (Please see page 1, lines 40 in the revised manuscript). The revised manuscript is re-written by someone with a better command of the English language.

The study authors fail to make clear what the rationale is of this study. Why is it important to investigate factors that are associated with TCM use in CTS? Is there any evidence in favor of TCM as a CTS treatment? Is it desirable that many people use TCM for the treatment of CTS or should they ideally prefer other more evidence-based treatments? The statement ‘The purpose of this study is to clarify the demographics and patterns of TCM used for CTS through the National Health Insurance (NHI) Database in Taiwan.’ is too vague. What do you mean by clarifying demographics and patterns, which factors would you like to study and why?

Reply: Thank you for the comments and suggestions. We make clear the rationale for this study that “CTS can relieve pain through non-steroidal anti-inflammatory drugs (NSAIDs), physical therapy or surgery. However, NSAIDs may cause kidney damage; ultrasound may have nerve damage concerns and surgery may injure normal tissues. Therefore, it is necessary to seek alternative therapy for CTS. Seradge et al. (2002) investigated the relation between CTS and alternative medicine treatment, and presented alternative medicine treatment can significantly save human suffering, medical costs, lost work time and socioeconomic problems [8].” in the revised manuscript (Please see page 2, lines 72-77 in the revised manuscript). We also state a clear study objective that “However, the main medical benefit of health care for TCM outpatient services come from the universal health insurance (NHI) program, in which the total amount of medical expenses assigned to TCM accounted for merely about 3%. We wonder whether such a distribution of payments is sufficient to provide safe and effective medical services. Among populations with CTS, we investigated factors associated with TCM use, including demographics, socioeconomic status, medical conditions and healthcare utilizations by using the nationwide representative claims data. This information would contribute to the medical resource allocation and develop the policy of integrative health services for CTS populations.” in the revised manuscript (Please see page 2, lines 88-95 in the revised manuscript).

You are recommended to use the STROBE guidelines for reporting cross-sectional studies. What is meant by epidemiological history assessment? A large number of demographic variables and comorbidity variables have been listed. Are these variables all variables of interest or are some of them considered confounders? Please clarify? I can imagine that there might be some clustering of data. For example, in urban areas and geographic regions. Did you consider multi-level logistic regression analyses or did you have reasons to stick to normal logistic regression analyses?

Reply: Thank you for the comments and suggestions. For avoiding confusion, we re-write “epidemiological and medical history assessments” to “medical history assessments” in the revised manuscript (Please see page 3, lines 120 in the revised manuscript). The variables listed in the study include the influence factors and confounders might be related to TCM use in CTS patients (Please see page 4, lines 128-138 in the revised manuscript). We add the multi-level logistic regression analyses in the revised manuscript (Please see page 5, lines 167-168 and page 8, Table 3 in the revised manuscript).

Discussion and conclusions. The female lead to a higher probability of CTS is due to the long-term fixed posture of the work patterns.’ This is speculative. This wasn’t investigated in this study. At least some some solid references are need to substantiate this statement. Otherwise, it should be removed. “Although the number of people living in urban areas use more proportion of TCM, however, people living in rural areas have a relatively high probability of using TCM to treat CTS.” I don’t understand this. This is a contradiction.

Reply: We agree with your suggestions and deleted the statement “The female lead to a higher probability of CTS is due to the long-term fixed posture of the work patterns.” in the revised manuscript (Please see page 10, lines 266 in the revised manuscript). We also re-state a clear study conclusion that “Among CTS patients, female sex, NHI registrations in central, southern and eastern Taiwan, as well as urbanization level with rural were associated with initiation of TCM use.” in the revised manuscript (Please see page 10, lines 267-269 in the revised manuscript).

Reviewer 3 Report

In Data source: “a population of 1 million subjects randomly selected from the insured 23.75 million people between 1996 and 2000”, please describe the random method. In Table 1:“52 90.21” should be changed to “52 (0.21)”;

                           “51548 (19.8)” should be changed to “5148 (19.8)”;

                           “17108 (67.7)” should be changed to “17108 (65.9)”;

                           “17174 66.1)” should be changed to “17174 (66.1)”;

                           “4456 917.2)” should be changed to “4456 (17.2)”;

                           Should “No. (%)” be changed to “n (%)”?

                           In number of inpatient visit, the P value of median is 0.009, please determine if it is correct?

                           Please mark the meaning of Q1, Q2, Q3.

                           The mean (SD) of age is measurement data, independent sample T test should be used for comparison between groups. And please explain the statistical analysis method of measurement data in “Statistical analysis” part.

In Table 2:Should “No. (%)” be changed to “n (%)”? In Table 1 and Table 2: It is recommended that all the percentages retain the same number of decimal places.

Author Response

Responses to reviewer’s comments

Ref.: MS ID#: ijerph-591433

MS TITLE:

The authors thank the reviewers for their helpful and thoughtful comments and suggestions. In the document below we have listed our responses point by point. Reviewer’s comments are numbered and in Cambria font. Responses follow in Calibri font.

Response to Reviewer

In Data source: “a population of 1 million subjects randomly selected from the insured 23.75 million people between 1996 and 2000”, please describe the random method.

Reply: Thank you for the comment and suggestion. We describe the random method that “The random number function, Sun Work Shop C 5.0, was used to generate random numbers and linear congruential random number generation was further used to select 1.1 million random numbers, from which the repetitive random numbers (about 20,000 to 30,000) were removed. Finally, the remaining numbers were divided into 20 times. Each time, 50,000 pieces of data were taken out in sequence, and a total of 1 million random numbers were obtained.” in the revised manuscript. (Please see page 3, lines 104-109 in the revised manuscript).

In Table 1:“52 90.21 should be changed to 52 (0.21); “51548 (19.8)” should be changed to “5148 (19.8)”; “17108 (67.7)” should be changed to “17108 (65.9)”; “17174 66.1)” should be changed to “17174 (66.1)”; “4456 917.2)” should be changed to “4456 (17.2)”; Should “No. (%)” be changed to “n (%)”? In number of inpatient visit, the P value of median is 0.009, please determine if it is correct? Please mark the meaning of Q1, Q2, Q3.

Reply: Thank you for your help. Your suggestions really help us improve the manuscript. We make revised based on your suggestions and collect the place of P value in number of inpatient visit (Please see pages 5-7, Table 1 in the revised manuscript).

The mean (SD) of age is measurement data, independent sample T test should be used for comparison between groups. And please explain the statistical analysis method of measurement data in “Statistical analysis” part.

Reply: Thank you for the comment and suggestion. We collect the statement that “These variables were compared between patients with carpal tunnel syndrome who used and did not use TCM as the first option by using chi-square tests in addition to the mean of age by using independent sample T test.” in the revised manuscript (Please see page 5, lines 155-157 in the revised manuscript).

In Table 2:Should (%) be changed to n (%)?

Reply: Thank you for your help. We make revised based on your suggestions (Please see page 7, Table 2 in the revised manuscript).

In Table 1 and Table 2: It is recommended that all the percentages retain the same number of decimal places.

Reply: Thank you for your help. We make revised based on your suggestions (Please see pages 5-7, Tables 1 and 2 in the revised manuscript).

Round 2

Reviewer 2 Report

Thank your for the revised paper. It has been improved substantially. Some statements in the introduction paragraph need appropriate referencing. It is stated that NSAIDs may cause kidney disease, ultrasound may damage the nerves and that surgery may injure normal tissues. Further substantiation of these statements is definitely required. 

The study authors also refer to a paper by Seradge et al (2002) on the outcomes of treatment for CTS but that does not answer whether or not TCM is an evidence-based treatment. Are there any Randomized trials in favor of CTM in this patient-group or is it based on other study designs. If not this should be stated as well since this is relevant information.

Author Response

Thank you very much for your comments and suggestions. We have revised the manuscript according to your suggestions (Please see the revised manuscript, page 2, paragraph 4). The revised manuscript has been checked and improved by a native English speaker.
